# The Progressive Utilization of Ponkan Peel Residue for Regulating Human Gut Microbiota through Sequential Extraction and Modification of Its Dietary Fibers

**DOI:** 10.3390/foods12224148

**Published:** 2023-11-16

**Authors:** Pu Gao, Meiyu Zheng, Hanyu Lu, Shengmin Lu

**Affiliations:** 1State Key Laboratory for Managing Biotic and Chemical Threats to the Quality and Safety of Agro-Products, Zhejiang Provincial Key Laboratory of Fruit and Vegetables Postharvest and Processing Technology, Ministry of Agriculture and Rural Affairs Key Laboratory of Post-Harvest Handling of Fruits, Institute of Food Science, Zhejiang Academy of Agricultural Sciences, Hangzhou 310021, China; gaopu2021@163.com (P.G.); zhenmey@sina.com (M.Z.); luhanyu1234@163.com (H.L.); 2College of Food Science and Technology, Nanjing Agricultural University, Nanjing 210095, China

**Keywords:** ponkan peel residue, dietary fiber, physical-enzymatic modification, short-chain fatty acids, gut microbiota, microbial diversity

## Abstract

As a by-product of citrus processing, ponkan (*Citrus reticulata* Blanco, cv. Ponkan) peel residue is a source of high quality dietary fiber (DF). To make a full utilization of this resource and give a better understanding on the probiotic function of its DF, soluble dietary fiber (SDF) and insoluble dietary fiber (IDF) were extracted from ponkan peel residue (after flavonoids were extracted) using an alkaline method, followed by modifications using a composite physical-enzymatic treatment. The in vitro fermentation properties of the modified SDF and IDF (namely, MSDF and MIDF) and their effects on short-chain fatty acids (SCFA) production and changes in the composition of human gut microbiota were investigated. Results showed that MSDF and MIDF both significantly lowered the pH value and enhanced total SCFA content in the broths after fermented for 24 h by fecal inocula (*p* < 0.05) with better effects found in MSDF. Both MSDF and MIDF significantly reduced the diversity, with more in the latter than the former, and influenced the composition of human gut microbiota, especially increasing the relative abundance of Bacteroidetes and decreasing the ratio of Firmicutes to Bacteroidetes (F/B) value. The more influential microbiota by MSDF were *g*-Collinsella, *p*-Actinobacteria and *g*-Dialister, while those by MIDF were *f*-Veillonellaceae, *c*-Negativicutes and *f*-Prevotellacese. These results suggested that the modified ponkan peel residue DF can be utilized by specific bacteria in the human gut as a good source of fermentable fiber, providing a basis for the exploitation of the citrus by-product.

## 1. Introduction

Ponkan (*Citrus reticulata* Blanco) originated in Asia, and its production has increased considerably since the early 21st century, with an increase of 63.73% between 2001 and 2018, which is concentrated mainly in China, Spain, Turkey, Morocco, Egypt and Brazil, accounting for 75.42% of total global production in 2018 (25.94 million metric tons) [1]. Citrus fruits for processed products such as cans, juices and jellies account for about 35% to 40% of total amount of the fruit yield in the world, producing peel residues in around 50% fresh weight of the fruits processed [2]. However, citrus by-products are often discarded, causing serious resource waste and environmental issues, or only processed into animal feed with low added value [3]. Citrus peel has many kinds of active ingredients, such as dietary fiber (DF), essential oils, flavonoids and pigment, which have various physiological effects such as anti-diabetic, anti-bacterial, anti-cancer and antioxidant effects [4,5]. DF, as a seventh nutrient, accounts for more than 50% of the weight of citrus peel residues, and has good hypoglycaemic and adsorptive properties [6]. Therefore, the rational utilization of DF resources in citrus peel residues will contribute to the sustainable development of citrus industry and health management, as well as a relief of environmental problems.

DF is a polysaccharide that is not digested and absorbed by the mammalian gastrointestinal tract, and usually requires reaching the colon to be partially or completely decomposed by colonic microorganisms, thus having advantages on improving intestinal microecology [7]. Previous studies [7,8,9] have reported that gut microbes could use DF to produce short-chain fatty acids (SCFA), including acetic, propionic and butyric acids, thus regulating pH of the gut, promoting the growth of beneficial groups of bacteria in the gut microbiota, and preventing potential colonization for pathogens through the competition for nutrients and adhesion sites. Additionally, ingestion of DF could reduce the risk of colon-related diseases and some metabolic syndromes such as obesity, diabetes, chronic kidney disease and systemic inflammation. DF from brans of cereals have been reported to modulate specific bacteria and promote the production of SCFA. For example, DF promoted the growth of butyrate-producing bacteria, including *Firmicutes*, *Dorea*, *Ruminococcus* and *Lachnospiraceae*; meanwhile, its metabolites stimulated the growth of related bacteria to enhance the intestinal barrier and modulate the immune function [10]. Degradation of DF from bagasse and passion fruit peels by human gut microbiota produced more SCFA and a more diverse bacterial community, demonstrating that fruit by-products could be used for selective regulation of human gut microbiota [11].

However, DF from different sources has varied compositions, ratios of soluble DFs (SDFs) to insoluble ones (IDFs) and functional properties. To enhance its some particular functional activities, DF before (in raw material) or after extraction is usually subjected to modification through chemical, physical or biological methods [12]. One common composite modification method is the combination of physical and enzymatic treatment, which has been shown to be effective in disrupting cell wall structure, promoting the conversion of IDF to SDF, and altering its physicochemical properties. Microwave plus enzymatic modification of grapefruit peels resulted in increased SDF content and enhanced functional properties, including water-holding, oil-holding, stability and adsorption properties [13]. Oats and wheat brans treated with superheated steam were reported to positively influence gut microbiota composition, leading to a significant increase in the relative abundance of lactic acid bacteria, *Rumenococci* and *Actinomycetes*, and a significant decrease in *Shigella* compared to those untreated [14]. In addition, previous studies have demonstrated that microwaves can be applied to improve the functional properties of DF. For example, the use of microwaves to modify DF from sweet potato pulp, apple pomace and soya bean residue resulted in increased SDF, enhanced SCFA production and significant changes to gut microbial communities of healthy adults [15]. A modification with cellulase and xylanase to DF from potato residues increased its SDF content, promoted the production of SCFA, and increased the abundance and diversity of gut microbiota in mice [16].

In this study, ponkan peel residues after flavonoids extracted beforehand with an ethanol solution were applied to extract DFs (SDF and IDF) using sodium hydroxide, which were subsequently modified by physical-enzymatic treatment. Our previous study showed the above method of extraction and modification increased the SDF content and improved their physicochemical properties such as water-holding, oil-holding, adsorption properties, etc. (submitted to a Chinese periodical). Therefore, to verify the regulation role of the modified soluble dietary fiber (MSDF) and modified insoluble dietary fiber (MIDF) in human gut microbiota, the pH, SCFA production and microbial community diversity in the fermentation broths after fecal microbiota transplantation were determined and assessed. The results would give a better understanding of the digestibility and fermentability of modified ponkan peel DF and their gut microbiota regulatory ability, thus offering a basis for exploiting ponkan peel residue or its modified DF as a potential functional ingredient in food industry.

## 2. Materials and Methods

### 2.1. Materials and Agents

Ponkan fruits were harvested in December 2022 in Quzhou city, Zhejiang province, China. Metaphosphoric acid, crotonic acid, acetic acid, propionic acid, isobutyric acid, butyric acid, isovaleric acid and valeric acid (all of chromatographic grade) were purchased from Sigma-Aldrich Trading Co., Ltd. (Shanghai, China). Cellulase, xylanase and inulin were obtained from Shanghai Yuanye Biotechnology Co., Ltd. (Shanghai, China).

### 2.2. Preparation of MSDF and MIDF from Ponkan Peel Residue

Fruit skin was manually peeled off ponkan fruits and freeze-dried. Dried peel was ground with a traditional Chinese medicine grinder (YD-150, Yongkan Sufeng Industry & Trade Co., Ltd., Jinhua, China) and screened by a 40-mesh sieve. DF was extracted using an alkali aqueous solution (pH 10.0) from the peel residue whose flavonoids were extracted in advance using 60% ethanol [17]. The extracted DF was modified using a physical-enzymatic method as described as follows [18]. The DF was mixed with PBS buffer (pH 5.9) in a ratio of 1:30 (*w*/*v*), and the suspension was run through a high-speed homogenizer (FSH-2A, New Rui Instrument Factory, West Jintan District, Changzhou, China) at 40,000× *g* (20,000 r/min, 22 to 25 °C) for 10 min. Two percent complex enzymes (cellulase: xylanase = 2:1) of the DF dry weight (*m*/*m*) was added directly to the homogenized suspension for hydrolysis at 50 °C for 1 h, and then the hydrolysis solution was centrifuged at 9600× *g* (4800 r/min, MC-4/7S, Qun An Experimental Equipment Co., Ltd., Ningbo, China) at 4 °C for 20 min. The precipitation was oven dried at 60 °C to obtain the modified IDF (namely, MIDF). The supernatant was precipitated with 4 folds volume of 95% ethanol and then centrifuged at 9600× *g* (4800 r/min) for 20 min. The precipitate was oven dried at 60 °C to obtain the modified SDF (namely, MSDF). MTDF amount was calculated by adding MIDF and MSDF together.

### 2.3. Analysis of Chemical Compositions of Ponkan Peel Residue and Its Modified Dietary Fibers

Moisture content was measured by the 105 °C oven method with reference to GB 5009.3 [19]. Ash content was determined by combustion method according to GB 5009.4 [20]. Protein and fat contents were tested by Kjeldahl and Soxhlet extraction method following to GB 5009.5 [21] and GB 5009.6 [22], respectively. Dietary fiber content was determined according to GB 5009.88 [23].

### 2.4. In Vitro Fermentation of Modified Dietary Fibers

#### 2.4.1. Preparation of Fecal Inocula

Six healthy volunteers (three males and three females, aged between 22 and 26 years) were selected to collect their fresh feces, and they had not taken therapeutic antibiotics during the past three months. Fecal samples were collected and weighed, transferred to an anaerobic chamber, diluted with PBS buffer (pH 5.9) containing 1% L-cysteine hydrochloride at a ratio of 1:5 (*w*/*v*), vortexed and left to stand for 5 min, then filtered through four layers of sterile gauze to obtain the filtrate. Equal amounts of fecal filtrate from the six volunteers were mixed.

#### 2.4.2. Preparation of Fermentation Media

The basal medium was prepared according to the method described by Schwab et al. [24]. Briefly, 5.0 g peptone, 4.5 g yeast extract, 0.5 g bile salts, 0.1 g NaCl, 2.0 g NaHCO_3_, 0.5 g L-cysteine hydrochloride, 0.04 g KH_2_PO_4_, 0.04 g K_2_HPO_4_, 0.01 g MgSO_4_-7H_2_O, 0.01 g CaCl_2_-6H_2_O, 0.02 g hemoglobin chloride, 2.0 g Tween 80, 10 μL vitamin K_1_ and 1.0 mL resazurin solution (1.0%, *w*/*v*) were mixed to form 1.0 L of fermentation medium using sterile water as solvent. The pH was adjusted to 7.0 with 0.1 mol/L HCl, and the medium was sterilized at 121 °C for 15 min for later use.

#### 2.4.3. In Vitro Fermentation Process

In an anaerobic environment, DF samples were sterilized by exposure to UV light for 2 h and their mass fraction added to the fermentation medium was 0.5% (*m*/*v*). The mixture (5 mL) was put in a Hungate anaerobic tube sterilized beforehand and then transferred to an anaerobic incubator (SW-II, Shengwei Electronic Technology Co., Shanghai, China). Mixed fecal filtrate was added to the Hungate anaerobic tube at 2% (*v*/*v*) using a 1 mL syringe and incubated in the anaerobic incubator at 37 °C with shaking at 600× *g* (300 r/min). At the intervals of 0, 6, 12 and 24 h, 0.5 mL of the fermentation broth was taken out from the anaerobic tube, rapidly transferred to a sterilized tube and stored in a −80 °C refrigerator. Inulin was used as a positive control (AC) and no sample added was regarded as a blank control (CK). The fermentation was divided into four groups, namely, MSDF, MIDF, AC and CK groups. Three parallel experiments were performed in each group.

#### 2.4.4. Determination of pH and SCFA Contents in Fermentation Broths

The pH of fermentation broth was measured using a pH meter (PHB-4, Shanghai Oshitol Industrial Co., Shanghai, China). Fecal fermentation broth (1.5 mL) was centrifuged (9000× *g*, 5 min, 4 °C) and 0.5 mL of the supernatant was taken out and put into a 1.5 mL sterilized centrifuge tube and 0.1 mL of crotonic acid metaphosphate solution was added. The resulting solution was frozen at −80 °C for 24 h, then thawed and centrifuged (9000× *g*, 5 min, 4 °C). The supernatant was collected and filtered through a 0.22 μm organic filtration membrane. The filtrate (100 μL) was then added to the inner tube of the gas phase sampling bottle. The contents of SCFA were determined by gas chromatography with a GC2010 plus instrument (Shimadzu Corp., Kyoto, Japan). A flame ionization detector with nitrogen carrier gas was employed for detection at injection volume, inlet temperature, detector temperature and split ratio of 1.0 μL, 220 °C, 250 °C and 8:1, respectively [25]. The column used was a DB-EEAP one (30.0 m × 0.32 mm × 0.50 μm, Agilent, Santa Clara, CA, USA). Six SCFA standards were used for identification and quantification. After detection, the SCFA content was calculated using the standard curve.

#### 2.4.5. Analysis of Gut Microbiota

Genomic DNA was extracted from the fermentation broth sediment using the Tianan Genetics Kit (Tiangen Biotech. Co., Ltd., Beijing, China) according to the kit’s instructions. The V4 to V5 regions of the bacterial 16S rDNA gene were amplified by polymerase chain reaction (PCR) using 515F and 907R primers for specific DNA fragments. PCR products were examined on an agarose gel and purified by the AxyPrep DNA Gel Extraction Kit (Axygen Biosciences, Union City, CA, USA). Purified PCR products were collected and aligned on the Illumina Miseq platform (Illumina Inc., San Diego, CA, USA). Using Silva as the reference database, the feature sequences were taxonomically annotated using a plain Bayesian classifier to obtain the species’ taxonomic information corresponding to each feature, and then species’ abundance, alpha diversity and beta diversity were computed and exhibited using Qiime2 software (Quantitative Insights into Microbial Ecology, v1.8.0, https://qiime.org, accessed on 15 March 2023). The online platform BMKCloud (https://www.biocloud.net, accessed on 15 March 2023) was used to analyze the sequencing data.

### 2.5. Statistical Analysis

Data in triplicate experiments were presented as means ± standard error and analyzed using SPSS 26.0 software (Chicago, IL, USA). The *p* < 0.05 was considered statistically significant. Moreover, the figures were depicted by Origin 2021 software (OriginLab, Northampton, MA, USA).

## 3. Results and Discussion

### 3.1. Chemical Compositions of Ponkan Peel Residue and Its Modified Dietary Fibers

The compositions of ponkan peel residue (PPR) and its modified dietary fiber (MDF) are shown in Table 1. Compared to PPR, MDF possessed significantly increased total dietary fiber (TDF) from 66.89 ± 2.60% to 89.59 ± 0.79% and SDF from 8.51 ± 0.65% to 28.72 ± 0.83% (*p* < 0.05). However, the variation in their IDF mass fractions was not significant (*p* > 0.05). This phenomenon was due to the high speed homogenization involved in shear and pressure disruption of the samples, resulting in looser spatial structure of dietary fibers, thus allowing more binding sites for enzymatic processing and making enzymatic hydrolysis more effective in disrupting the structure [26]. The changes in moisture, ash, protein and crude fat mass fractions of MDF were significantly lower compared to those of PPR (*p* < 0.05). This was because the purity of dietary fiber in MDF was increased as a result of the physical enzymatic modification process, which broke down the large molecules into smaller ones, and their loss in the solvent. The results indicated that physical-enzymatic composite modification was significantly favorable in improving SDF content in ponkan peel DF.

### 3.2. Effect of Modified Dietary Fibers on pH of In Vitro Fermentation Broths

During the fermentation of substrates by the human intestinal microbiota, a number of acidic products can be formed, including lactic acid and SCFA, which therefore affect the pH and microbial diversity in intestinal tract [27]. Therefore, pH is a very important indicator to reflect the in vitro fermentation process. Figure 1 shows the change in pH during the in vitro fermentation of MDF and the controls. The pH of the fermentation broths in CK, AC, MSDF and MIDF groups all decreased gradually from the initial 7.0 as the fermentation time prolonged. At the beginning of fermentation (0–6 h), the pH of all groups dropped significantly (*p* < 0.05), with the most in the AC and the least in the CK, and more in MSDF than MIDF. After 24 h of fermentation, the pH of CK, AC, MSDF and MIDF groups were 5.07, 4.32, 4.79 and 4.87, respectively. The decrease in pH of broths might be due to the consumption of carbohydrates in the medium by intestinal microorganisms, leading to fermentation and subsequent formation of SCFA, thus lowering the pH [28]. The result suggested that additions of MSDF and MIDF both help lower the pH in colon to some extent and maintain a balanced and acidic intestinal environment. A lower pH in colon helps promote the growth of probiotics and prevent the growth of harmful bacteria, thus ensuring a good intestinal microecology.

### 3.3. Effect of Modified Dietary Fibers on the SCFA Contents of In Vitro Fermentation Broths

Indigestible DF can act as a major source of energy and carbon for the gut microbiota, stimulating their growth while producing SCFA [29]. The concentration of SCFA is one of the most important indicators to assess the prebiotic activity of DF. Figure 2 shows the contents of acetic acid, propionic acid, butyric acid and total SCFA in broths among groups at the end of fermentation (24 h), with acetic acid being the highest, followed successively by propionic acid and butyric acid. Isobutyric acid, valeric acid and isovaleric acid were present at low levels and their contents were not directly displayed in the figure. The acetic acid concentrations in MSDF and MIDF treated groups (23.85 and 21.52 mmol/L) were significantly higher than that in the CK (*p* < 0.05). Propionic acid and butyric acid concentrations (6.17 mmol/L and 1.30 mmol/L in MSDF treated group and 5.20 mmol/L and 1.27 mmol/L in MIDF group) were also significantly higher than those in the CK (*p* < 0.05), but showed no significant difference compared to those in the AC. Previous studies have found that acetic acid helps inhibit the growth of intestinal pathogens, while propionic acid may affect the liver and cholesterol metabolism, and butyric acid may be used as a fuel for intestinal cells [29]. At the same time, acetic and propionic acids not only provide energy to the liver and surrounding tissues, but also play an important role in gluconeogenesis and lipogenesis [30]. Total SCFA concentrations were 31.45 mmol/L and 28.10 mmol/L in MSDF and MIDF treated groups, respectively, both higher than that in the CK (23.34 mmol/L, *p* < 0.05), but there was no significant difference between MSDF group and the AC (32.71 mmol/L), which was similar to the results of previous studies [9]. Our results indicated that both ponkan peel residue MSDF and MIDF as fermentation substrates well promoted the production of SCFA by human intestinal flora, with better performance found in MSDF.

### 3.4. Effect of Modified Dietary Fibers on Gut Microbiota of In Vitro Fermentation Broths

#### 3.4.1. Alpha Diversity and β Diversity

Alpha (α) diversity can reflect the abundance and diversity of the microbial community within a specific ecosystem, including Chao, Ace, Shannon and Simpson indices. The Chao and Ace indices are commonly used in ecology to estimate the total numbers of species and can be used to reflect community richness, with a higher index indicating a richer microbial community [31]. The Shannon and Simpson indices are used to assess the diversity of microbial communities, with higher values indicating higher and lower community diversity, respectively [32]. The differences in Chao and Ace indices among the CK, AC, MSDF and MIDF groups were not statistically significant (Figure 3a,b), suggesting that the addition of substrate has little effect on the abundance of the microbial community after fermentation. As seen from Figure 3c,d, the Shannon and Simpson indexes of microbial communities in the MSDF and MIDF groups were significantly lower and higher than those of the CK and AC (*p* < 0.05), respectively, indicating that the diversity of the gut microbiota was reduced after in vitro fermentation with the addition of MSDF and MIDF as substrates. This could be attributed to the presence of MDF, which potentially inhibited the growth of specific harmful microorganisms during the fermentation process. Moreover, the reduction in diversity of gut microbiota was significant between MSDF and MIDF groups with more in the latter than the former (*p* < 0.05).

Beta (β) diversity is often used to highlight variation in diversity of gut microbiota between samples, and the closer the samples to each other, the more similar the composition of the gut microbes [33]. In order to assess and determine the differences in taxonomic operational units (OTU) in the samples, principal coordinate analysis (PCoA) and non-metric multidimensional scaling analysis (NMDS) were performed. The results of PCoA are shown in Figure 4a, with principal component (PC) 1 and 2 accounting for 63.71% and 28.99% of the variance, respectively. It was seen that the data from the same sample (r = 3) were close to each other, indicating that the data were relatively uniform within the same group. However, the distance between the CK and AC groups and the MSDF and MIDF groups was relatively long, indicating great difference in their effects on gut microbial species, but a very close distance was found between MSDF and MIDF groups, suggesting their high similarity in affecting gut microbiota. NMDS focused on assessing the similarity in gut microbiota diversity among different groups, and its results showed that the MSDF and MIDF groups were far from the CK and AC groups, and the MSDF group was very close to MIDF one, indicating their higher similarity in influencing gut microbiota diversity (Figure 4b). This was consistent with the PCoA results. The above results suggested that MSDF and MIDF have significant but similar influence on diversity of human gut microbiota.

#### 3.4.2. Composition of Gut Microbiota

Figure 5a shows that after in vitro fermentation, 95% of the gut microbiota at the phylum level consist of *Firmicutes*, *Bacteroidetes*, *Proteobacteria* and *Actinobacteria*. The CK group contained a small number of *Fusobacteria*, under which the survival of *Clostridium* in the human intestine would lead to an abnormally active state in colon cancer cells and was associated with a high incidence of colon cancer [34]. In contrast, the relative abundance of *Fusobacteria* in the sample groups was extremely low, which suggested that the addition of MSDF or MIDF or AC potentially inhibit the incidence of colon cancer. As seen in Figure 5a, the fermentation after addition of MSDF and MIDF resulted in a decrease in the relative abundance of *Firmicutes* and an increase in that of *Bacteroidetes* compared to CK, which might have led to a decrease in energy intake. It has been made known that the ratio of relative abundance in *Firmicutes* to *Bacteroidetes* (F/B) is an important index of weight loss, and a lower F/B value indicates that the fermentation substrate has a weight loss function [35]. One of the main producers of propionic acid in the colon is *Bacteroidetes*, which regulates blood lipids synthesis and cholesterol level rise. The decreased F/B value corresponded to significantly increased propionic acid levels in the MSDF and MIDF groups compared to the CK. In addition, the addition of MSDF and MIDF to gut microbiota reduced the relative abundance of *Proteobacteria*. It has been shown that ecological dysregulation of microbiota during metabolic disorders usually includes an increase in the relative abundance of the *Proteobacteria* [36]. The results suggested that MSDF and MIDF can help maintain gut ecological balance and exert potential wholesome function. Furthermore, MDF (especially MSDF) significantly increased the relative abundance of *Actinobacteria*, a group of microorganisms producing antibiotics and enzymes and playing an important role in maintaining intestinal health. Such conclusion had also been demonstrated by Dominianni et al. [37], who found that DFs from soy, vegetables and fruits all increased the relative abundance of actinomycetes, which was essential for health of gastrointestinal tract and wellness of whole organism.

The relative abundance of microbial composition at the genus level in each group of fermentation broths (Figure 5b) indicated that *Escherichia*-*Shigella*, *Megasphaera*, *Prevotella*, *Lactobacillus*, *Collinsella*, *Megamonas*, *Bacteroides*, *Streptococcus* and *Enterococcus* were the dominant genera of the gut microbiota. *Prevotella* helps to break down proteins and carbohydrates, and intake propionic acid from arabinoxylan and oligofructose, which lowers serum cholesterol and reduces liver fat production [38]. *Megamonas* also ferments various carbohydrates, with the end products being acetic acid, propionic acid and lactic acid. *Lactobacillus* is a beneficial microorganism that ferments polysaccharides to produce lactic acid and is now widely recognized for its role in maintaining human health and regulating immune function [39]. However, *Enterococcus* is by far one of the most important pathogens of hospital-acquired infections and most often causes urinary tract infections. As seen in Figure 5b, MSDF and MIDF promoted the growth of beneficial intestinal bacteria such as *Prevotella*, *Megamonas* and *Lactobacillus* in enhanced abundance. In addition, the relative abundance of *Bacteroides*, *Escherichia-Shigella* and *Enterococcus* in the MSDF and MIDF groups decreased after fermentation, suggesting that the addition of MSDF and MIDF inhibit the growth of some of the harmful bacteria.

Linear discriminant analysis effect size (LEfSe) and latent Dirichlet allocation (LDA) can be combined for more specific identification of microbial community members. The results of LEfSe analysis on gut microbiota affected by MDF are shown in Figure 6a, with each level of taxonomy from phylum to genus presenting separately from the inside to outside, and the size of the nodes and area representing the average relative abundance of species. Yellow nodes indicate species that are not significantly different, and colorful ones in each group represent those having a significant effect on differences between groups. A bar chart of the LDA discrimination of the gut microbiota, based on LDA > 2 and *p* < 0.05, is shown in Figure 6b, demonstrating species with significant differences in abundance among different groups and the bar lengths reflecting the influence size of species with statistical differences. The analysis results indicated that four genera, including *o-Clostridiales*, *f-Peptostreptococcaceae*, *f-Bacteroidaceae* and *g-Peptoniphilus*, in the CK had a greater influence on the gut microbiota. The more influential microbiota in the MSDF group were *o-Coriobacteriales*, *g-Collinsella*, *p-Actinobacteria* and *g-Dialister*, while those in the MIDF group were *f-Veillonellaceae*, *c-Negativicutes*, and *f-Prevotellacese*. In the AC group, *c-Bacilli*, *o-Lactobacillales*, and *f-Streptococcaceae* were found to be more influential.

## 4. Conclusions

This study investigated the in vitro fermentation characteristics of ponkan peel residue’s modified dietary fibers (MSDF and MIDF) prepared with physical-enzymatic method and their effects on human gut microbiota. The DF was extracted from the peel residue whose flavonoids were extracted first. The results indicated that the addition of MSDF and MIDF both increased the production of acids, especially for MSDF, significantly lowering the pH in the fermentation broth, while increasing the production of SCFA (*p* < 0.05). In addition, MSDF and MIDF were fermented by specific microorganisms that regulated the composition of human intestinal microbiota to varying degrees. Of these, both MSDF and MIDF promoted the growth of beneficial bacteria, such as *Megasphaera*, *Prevotella* and *Collinsella*, and reduced the relative abundance of *Bacteroidetes*, *Escherichia-Shigella* and *Enterococcus*. The more influential microbiota by MSDF were *g*-*Collinsella*, *p-Actinobacteria* and *g-Dialister*, while those by MIDF were *f*-*Veillonellaceae*, *c*-*Negativicutes* and *f-Prevotellacese*. The results might draw the public’s attention to DF’s regulation role on intestinal microbiota, which will still need further validation on its probiotic functions through in vivo test. The study also provokes the thought of progressive utilization of citrus peel through sequential extraction or modification for developing wholesome dietary supplements.

## Figures and Tables

**Figure 1 foods-12-04148-f001:**
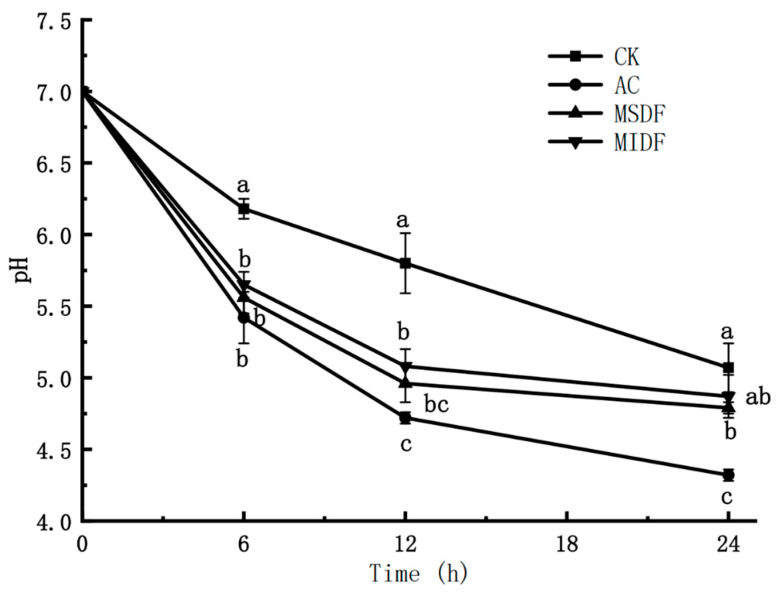
Changes in pH during in vitro fermentation of ponkan peel residue MDF. Different lowercase letters on value curves of groups at each sampling time represent significance at *p* < 0.05. CK, blank control; AC, positive control; MSDF, modified soluble dietary fiber; MIDF, modified insoluble dietary fiber.

**Figure 2 foods-12-04148-f002:**
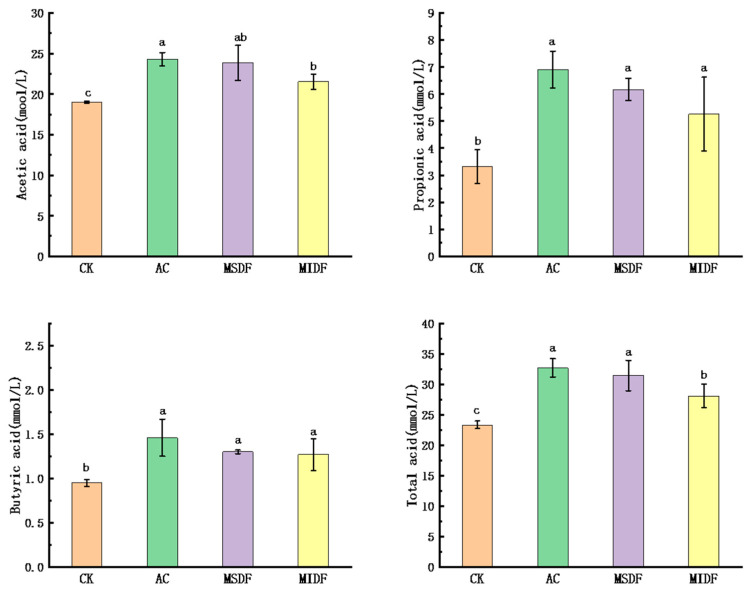
Changes in SCFA contents after in vitro fermentation of ponkan peel residue modified dietary fibers. Different lowercase letters on the tops of columns in each chart indicate significance in data among groups at *p* < 0.05. CK, blank control; AC, positive control; MSDF, modified soluble dietary fiber; MIDF, modified insoluble dietary fiber.

**Figure 3 foods-12-04148-f003:**
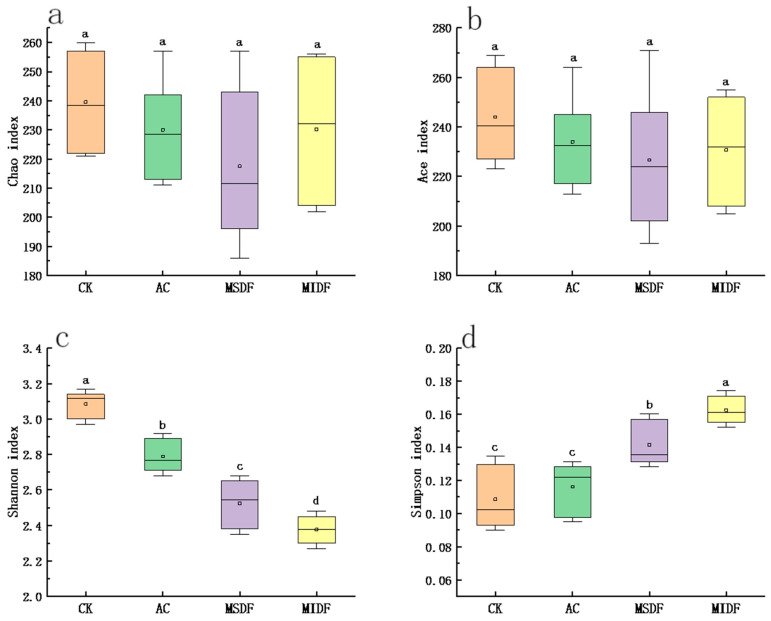
Changes in the alpha diversity of intestinal microorganisms after in vitro fermentation with ponkan peel residue MDF. (**a**) Chao index; (**b**) Ace index; (**c**) Shannon index; (**d**) Simpson index. Different lowercase letters on the tops of columns in each chart indicate significance in data among groups at *p* < 0.05. CK, blank control; AC, positive control; MSDF, modified soluble dietary fiber; MIDF, modified insoluble dietary fiber.

**Figure 4 foods-12-04148-f004:**
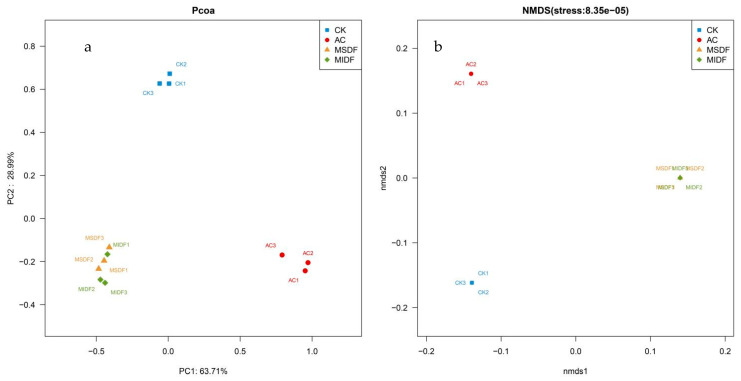
Plots of PCoA (**a**) and NMDS (**b**) analyses on gut microbes among in vitro fermentation broths of different samples. CK, blank control; AC, positive control; MSDF, modified soluble dietary fiber; MIDF, modified insoluble dietary fiber.

**Figure 5 foods-12-04148-f005:**
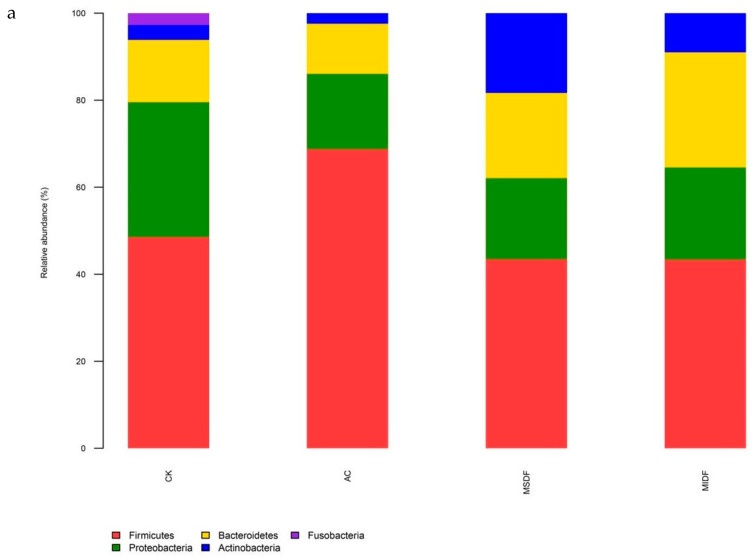
Composition difference of the gut microbiota at phylum (**a**) and genus (**b**) taxonomic level among in vitro fermentation broths of different samples. CK, blank control; AC, positive control; MSDF, modified soluble dietary fiber; MIDF, modified insoluble dietary fiber.

**Figure 6 foods-12-04148-f006:**
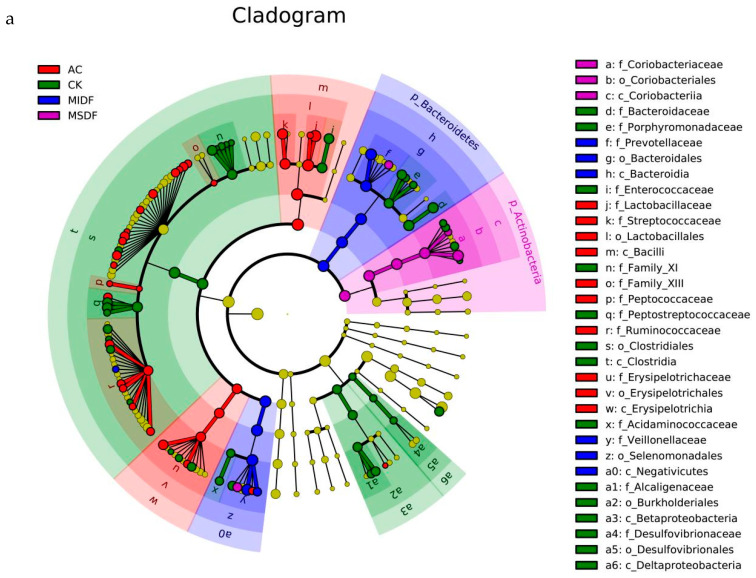
Plots of LEfSe (**a**) and LDA (**b**) analysis on gut microbiota among in vitro fermentation broths of different samples. CK, blank control; AC, positive control; MSDF, modified soluble dietary fiber; MIDF, modified insoluble dietary fiber.

**Table 1 foods-12-04148-t001:** Basic chemical compositions of PPR and MDF.

Basic Composition (%)	PPR	MDF
Moisture	2.27 ± 0.19 a	1.10 ± 0.10 b
Ash	8.41 ± 0.38 a	5.57 ± 0.22 b
Protein	6.89 ± 0.14 a	1.14 ± 0.09 b
Crude fat	2.20 ± 0.21 a	0.51 ± 0.09 b
TDF	66.89 ± 2.60 b	89.59 ± 0.79 a
IDF	58.38 ± 2.07 b	60.87 ± 0.85 b
SDF	8.51 ± 0.65 b	28.72 ± 0.83 a

Different lowercase letters after data in each row indicate significance at *p* < 0.05. PPR, ponkan peel residue; MDF, modified dietary fiber; TDF, total dietary fiber; IDF, insoluble dietary fiber; SDF, soluble dietary fiber.

## Data Availability

Data are available upon a request from the corresponding author and approved by the affiliations.

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
