# Peer review of "The Progressive Utilization of Ponkan Peel Residue for Regulating Human Gut Microbiota through Sequential Extraction and Modification of Its Dietary Fibers"

_foods, 2023, doi:10.3390/foods12224148_

Round 1

Reviewer 1 Report

Comments and Suggestions for Authors

The idea behind the scientific paper is good and interesting. The methodology used is also relevant. However, the main drawback of the paper lies in its conclusions and interpretation of the results indicating that the authors may not have fully considered the necessary steps to draw concrete scientific claims supporting the functionality of Ponkan peel DF. In drawing conclusions, it is essential to adjust them, considering that they stem from in vitro experiments that can only serve as a basis for further hypotheses, not as grounds for claims about bioactive/health promotional effects (see lines 308-310 for example).

All the Figures (especially Fig 6) are of low resolution, so the text on them is not visible/readable. Please provide higher-resolution Figures.

Editing of English language required.

Further detailed comments are provided below:

Line 34-36: Considering that the information you provided is from 2018, and therefore quite outdated, please either emphasize a trend from the last decade or provide more recent data.

line 47: value of citrus and health industry??? This sentence structure is very unclear; it sounds like there is a health industry?

117: precipitation or precipitant

All the Figures are of low resolution, so the text on them is not visible/readable. Please provide higher-resolution Figures

273-274: Sentence is not clear. Suppose the authors intended to say: This could be attributed to the presence of MDF, which potentially inhibited the growth of specific harmful microorganisms during the fermentation process.

308-310: Making such a claim without conducting studies for validation is quite bold. Please consider rephrasing the sentence.

381-382: Authors are encouraged to rephrase the conclusion, as it is not adequate for a scientific paper. The core of the research should focus on initiating further investigations to substantiate the assumptions made. These studies should aim to provide evidence regarding whether Ponkan peel residue truly exhibits the activity described in the in vitro experiments presented in this study.

Comments on the Quality of English Language

Editing of English language required.

Author Response

The idea behind the scientific paper is good and interesting. The methodology used is also relevant. However, the main drawback of the paper lies in its conclusions and interpretation of the results indicating that the authors may not have fully considered the necessary steps to draw concrete scientific claims supporting the functionality of Ponkan peel DF. In drawing conclusions, it is essential to adjust them, considering that they stem from in vitro experiments that can only serve as a basis for further hypotheses, not as grounds for claims about bioactive/health promotional effects (see lines 308-310 for example).

Reply: Thank you for your positive comments and pointing out review opinion on writing conclusions. We have accordingly adjusted the conclusions and done our best to interpret the results.

All the Figures (especially Fig 6) are of low resolution, so the text on them is not visible/readable. Please provide higher-resolution Figures.

Reply: Thank you for your opinion on figures’ resolution. We have replaced figures of higher resolution.

Editing of English language required.

Reply: We have done our best to edit the language of English throughout the manuscript.

Further detailed comments are provided below:

Line 34-36: Considering that the information you provided is from 2018, and therefore quite outdated, please either emphasize a trend from the last decade or provide more recent data.

Reply: Thank you for your comment. We have emphasized an increase trend from 2001 to 2018.

line 47: value of citrus and health industry??? This sentence structure is very unclear; it sounds like there is a health industry?

Reply: Thank you for your questions. We have revised this sentence to be more clear in expression.

117: precipitation or precipitant

Reply: Thank you for pointing out this mistake. The word should be precipitate (Noun).

All the Figures are of low resolution, so the text on them is not visible/readable. Please provide higher-resolution Figures

Reply: Thank you again. We have adjusted them.

273-274: Sentence is not clear. Suppose the authors intended to say: This could be attributed to the presence of MDF, which potentially inhibited the growth of specific harmful microorganisms during the fermentation process.

Reply: Thank you for your suggestion. We have revised accordingly.

308-310: Making such a claim without conducting studies for validation is quite bold. Please consider rephrasing the sentence.

Reply: Thank you for your comment and reminding. We have revised the conclusion accordingly.

381-382: Authors are encouraged to rephrase the conclusion, as it is not adequate for a scientific paper. The core of the research should focus on initiating further investigations to substantiate the assumptions made. These studies should aim to provide evidence regarding whether Ponkan peel residue truly exhibits the activity described in the in vitro experiments presented in this study.

Reply: Thank you for your comments and suggestions. We have revised accordingly.

Reviewer 2 Report

Comments and Suggestions for Authors

Thank you very much for manuscript entitled Progressive utilization of ponkan peel residue for regulating  human gut microbiota through sequential extraction and modification of its dietary fibers in Foods journal. Article is a topic of intrest. However some changes need to be addressed.

1. The English language throughout the manuscript need to be improved.

2.Apart from dietry fibers, why authors not focused phenolics in peel.??

3. Conclusion section need to be improved with some future illustrations.

4. Authors can merge figure 2 and 3.

5. More references can be included in the manuscript.

Comments on the Quality of English Language

Moderate

Author Response

Thank you very much for manuscript entitled Progressive utilization of ponkan peel residue for regulating human gut microbiota through sequential extraction and modification of its dietary fibers in Foods journal. Article is a topic of interest. However some changes need to be addressed.

  1. The English language throughout the manuscript need to be improved.

Reply: Thank you for your positive comments. We have done our best to revise the language of the manuscript.

  1. Apart from dietry fibers, why authors not focused phenolics in peel.??

Reply: Thank you for your question. As mentioned in Introduction and Materials and Methods parts, the peel used as a material for extracting dietary fibers was that had been firstly extracted for flavonoids (a kind of phenolics).

  1. Conclusion section need to be improved with some future illustrations.

Reply: We have revised the expression in the conclusion section.

  1. Authors can merge figure 2 and 3.

Reply: Thank you for your suggestion, however, we believe it is better when they are separate.

  1. More references can be included in the manuscript.

Reply: Thank you for your suggestion. We will include appropriate references if we find them.

Reviewer 3 Report

Comments and Suggestions for Authors

The manuscript is interesting and well perfomed. Only some minor questions should be adressed prior to its publication:

line 44: "DF" abbreviations should not be used at the beguining of the phrase

Line 49: "DF is a carbohydrate". Dietary fiber are not only carbohydrates, despite carbohydrates are its main component.

Lines 52-58. "Previous.....systemic inflammation" This phrase is too long to can be read as a unique phrase. Plase rewrite it.

Lines 66-67: This phrase seems not make sense. Please delete it.

Lines 123-124. The methods employed for proximate composition determination should be cited in the references list. It is not clear the meaning of "GB", because most often used methods are "ISO"

Line 145: "In vitro" should be written in italics

Line 80: Please insert an space between "Illumina" and "Miseq"

Table 1 and all other tables and figures: All the abbrevaition included in both Tables and Figures should be defined in the footnote. The table should be sense itshelf, indepedently of the main text.

Author Response

The manuscript is interesting and well perfomed. Only some minor questions should be adressed prior to its publication:

line 44: "DF" abbreviations should not be used at the beguining of the phrase

Reply: Thank you for your positive comments on our manuscript. We wonder the meaning of this question. We did use “DF” abbreviation for the first occurrence of the phrase “dietary fiber” either in Abstract or in Introduction section.

Line 49: "DF is a carbohydrate". Dietary fiber are not only carbohydrates, despite carbohydrates are its main component.

Reply: Thank you for your comment. We have revised carbohydrate into polysaccharide which is more correct.

Lines 52-58. "Previous.....systemic inflammation" This phrase is too long to can be read as a unique phrase. Plase rewrite it.

Reply: Thank you for your comment. The sentence is truly too long to read. We have rewritten it.

Lines 66-67: This phrase seems not make sense. Please delete it.

Reply: Thank you. We have rewritten this sentence.

Lines 123-124. The methods employed for proximate composition determination should be cited in the references list. It is not clear the meaning of "GB", because most often used methods are "ISO"

Reply: GB is abbreviation for national standard in Chinese. We have added the references.

Line 145: "In vitro" should be written in italics

Reply: Thank you for your reminding. We have corrected this.

Line 80: Please insert an space between "Illumina" and "Miseq"

Reply: Thank you for your kind reminding. Your strictness will help improve our manuscript’s level.

Table 1 and all other tables and figures: All the abbrevaition included in both Tables and Figures should be defined in the footnote. The table should be sense itshelf, indepedently of the main text.

Reply: Thank you for your reminding. We have added the annotation to abbreviation in both Tables and Figures in the footnotes.